# xMLP: Revolutionizing Private Inference with Exclusive Square Activation

## Abstract

Private Inference (PI) enables deep neural networks (DNNs) to work on private data without leaking sensitive information by exploiting cryptographic primitives such as multi-party computation (MPC) and homomorphic encryption (HE). However, the use of non-linear activations such as ReLU in DNNs can lead to impractically high PI latency in existing PI systems, as ReLU requires the use of costly MPC computations, such as Garbled Circuits. Since square activations can be processed by Beaver's triples hundreds of times faster compared to ReLU, they are more friendly to PI tasks, but using them leads to a notable drop in model accuracy. This paper starts by exploring the reason for such an accuracy drop after using square activations, and concludes that this is due to an "information compounding" effect. Leveraging this insight, we propose xMLP, a novel DNN architecture that uses square activations exclusively while maintaining parity in both accuracy and efficiency with ReLU-based DNNs. Our experiments on CIFAR-100 and ImageNet show that xMLP models consistently achieve better performance than ResNet models with fewer activation layers and parameters while maintaining consistent performance with its ReLU-based variants. Remarkably, when compared to state-of-the-art PI Models, xMLP demonstrates superior performance, achieving a 0.58% increase in accuracy with $7\times$ faster PI speed. Moreover, it delivers a significant accuracy improvement of 4.96% while maintaining the same PI latency. When offloading PI to the GPU, xMLP is up to $700\times$ faster than the previous state-of-the-art PI model with comparable accuracy.

## 1 Introduction

The growing demand for cloud-based deep learning services raises significant privacy concerns for both users and cloud service providers (Mishra et al., 2020), because input data from users may contain personal information while the trained deep neural networks' parameters may be trade secretes for cloud service providers. To alleviate such privacy concerns, *private inference* (PI) techniques based on cryptographic primitives, such as homomorphic encryption (HE) (Gentry, 2009) and multi-party computation (MPC) (Yao, 1982), have been proposed by Mishra et al. (2020); Liu et al. (2017); Juvekar et al. (2018) to protect both users and cloud service providers from leaking their respective sensitive information.

While PI's cryptographic primitives offer robust privacy guarantees, when applied to deep neural network (DNN) models, especially for nonlinear operations like ReLU, there is a notable performance challenge, resulting in significant latency. Recent works have been primarily focused on enhancing the efficiency of Private Inference (PI). Broadly speaking, these efforts fall into two main areas. First, there's the formulation of PI protocols, where the emphasis is on crafting faster algorithms to support nonlinear computations and streamlining the entire inference process to cut down on latency. Second, there's the design of neural networks which aims to maintain model performance while minimizing the use of computationally expensive operations like ReLU.

The first camp of PI research mostly focused on the PI protocol design tailored to specific DNN structures. GAZELLE (Juvekar et al., 2018), for instance, supports PI on neural networks consisting of both linear operations and nonlinear layers (ReLU). It employs optimized linearly-homomorphic encryption (LHE) (Brakerski et al., 2014; Fan & Vercauteren, 2012) for linear tasks and garbled circuits (GC) (Yao, 1986) for ReLU functions. To facilitate inter-layer communication, GAZELLE

incorporates additive secret sharing (Damgård et al., 2012). However, this method still incurs significant computational overhead, particularly when processing ReLU operations. This remains true even for more recent advancements, such as Delphi (Mishra et al., 2020).

To compensate for the computational overhead associated with nonlinear (ReLU) operations in PI, the second strand of research endeavors to reduce the quantity of ReLU operations within the network. For instance, DELPHI exemplifies this approach by introducing a more streamlined protocol than GAZELLE and modifying the DNN structure such as ResNet (He et al., 2016a), substituting some of the resource-intensive ReLU operations with more PI-friendly quadratic polynomial functions. Consequently, while DELPHI achieves greater computational efficiency, there's a trade-off in terms of some loss in model accuracy.

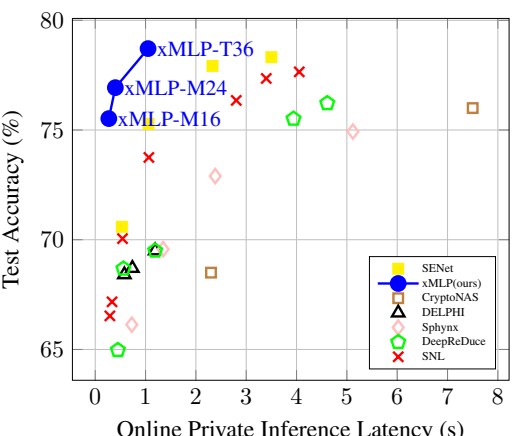

Figure 1: xMLP pushes the new Pareto frontier of Private Inference Latency vs. Accuracy on CIFAR-100.

Multiplication-based activation functions, such as quadratic polynomials, offer faster inference speeds because they can be efficiently computed using Beaver's multiplication triples (BT) (Beaver, 1991). However, substituting ReLU with these multiplication-based activation functions often results in a significant decline in the network's accuracy. This reduced accuracy might not be suitable for certain critical applications. Addressing this, DELPHI introduced a hybrid protocol supporting both ReLU and quadratic polynomials and employed a neural network architecture search (NAS) to identify where quadratic polynomials could replace ReLUs for improved PI performance. In a similar vein, CryptoNAS (Ghodsi et al., 2020) presents a refined NAS technique to discover the best network structure within a specified ReLU budget, while DeepReDuce (Jha et al., 2021) suggests more advanced ReLU reduction methods. Although all of these methods can reduce, to some degree, the removal of ReLU's impact on the accuracy loss, they still make the tradeoffs between model accuracy and PI performance, and there still exists a huge accuracy gap when compared to ReLU.

To address the aforementioned challenges, we present xMLP, an innovative architecture utilizing square activation functions. Remarkably, on the CIFAR-100 dataset, xMLP-M16 achieves a top-1 accuracy of 75.52% with just 2.2M parameters, outperforming ResNet-18 which reaches 75.43% with 11.4 million parameters. In PI, xMLP either achieves a 7× speedup or surpasses benchmarks by 4.96% in accuracy, redefining the state-of-the-art standards as shown in Figure 1. The main contributions of this paper are as follows:

- We propose, for the first time, a simple yet efficient neural network architecture that eliminates the need for ReLU activations, relying solely on quadratic functions. xMLP allows for faster PI while achieving performance on par with conventional DNNs. We empirically validate these claims on CIFAR-100, Tiny-ImageNet, and ImageNet datasets.

- We provide an analysis elucidating why multiplication-based activation functions have historically underperformed in neural networks, offering insights into their limitations.

- Leveraging the PI protocol from Delphi, we evaluate the PI performance of xMLP, comparing it with prior architectures. Our results indicate that xMLP sets a SOTA for PI in terms of both accuracy and efficiency.

## 2 BACKGROUND

**Private Inference (PI)** refers to a set of techniques that provide a strong privacy guarantee to both the data and deep learning model while preserving the model's functionality. In a typical PI pipeline, the user sends the encrypted data (e.g. secret shared) to the cloud, and the cloud runs inference

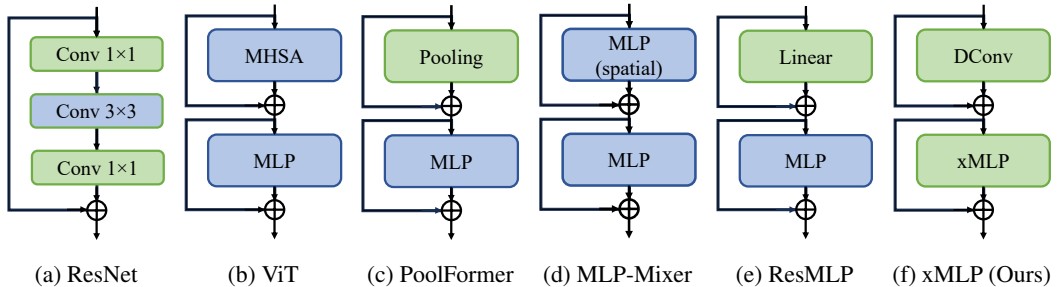

Figure 2: **DNN architectures.** "Green blocks" contain only PI-friendly operations (e.g., linear layers, square activation). "Blue" blocks contain non-PI- friendly operations (e.g., ReLU activation, softmax). "DConv" is the depth-wise convolution. Contrary to other architectures, xMLP includes only linear operations and square activation which are both PI-friendly. The details of our xMLP block are shown in Figure 4.

directly on the encrypted data without decrypting it, then sends the encrypted result back to the user. Depending on the cryptographic primitives used, recent PI systems fall into two types: (1) using only fully homomorphic encryption (FHE) (Gentry, 2009); (2) using both FHE and multi-party computation (MPC) (Yao, 1982) primitives. FHE-only PI systems leverage FHE algorithms such as CKKS (Cheon et al., 2017), have very high computation costs, and support only limited operations. Its advantage over other PI systems is that no extra communication cost between the user and the cloud is needed, i.e., all the computation is done on the cloud side. Hybrid PI systems (Mishra et al., 2020; Juvekar et al., 2018; Mohassel & Zhang, 2017) use homomorphic encryption (HE) for linear layers and MPC primitives for non-linear layers. These methods can incur large communication costs due to MPC primitives, but they have less computation cost, hence usually shorter overall inference latency. In this paper, we focus on the hybrid PI systems.

**Additive secret sharing (SS)** (Damgård et al., 2012) is defined as follows. For a given value $x$, SS divide $x$ into two secrete shares $[x]_1$ and $[x]_2$ so that $x = [x]_1 + [x]_2$. To reconstruct the secret value $[x]$, simply summing up secret shares from each party $[x]_1 + [x]_2 = x$. To calculate the sum of two secret shared values $[x]$ and $[y]$, each party $i \in \{1, 2\}$ will compute $[x]_i + [y]_i$.

**DELPHI** is a hybrid PI protocol that uses HE to process linear layers and uses MPC primitives to process non-linear layers. Specifically, DELPHI uses Beaver's multiplication triple (BT) (Beaver, 1991) to process quadratic activation layers and uses Garbled Circuits (GC) (Yao, 1982) to process ReLU layers. Among these operations, GC is the most costly one in terms of communication cost and inference latency. DELPHI consists of an offline phase and an online phase. Offline phase is performed once for every model, and online phase is performed for every user's input. During the offline phase, computations involving heavy HE computations that are independent of the user's input are pre-processed. The online phase then consists of applying linear layers to secret shared data and running MPC primitives for non-linear layers. We provide the details of DELPHL's protocol in the supplementary material.

**CrypTen** (Knott et al., 2021) is a PyTorch-based framework focused on Privacy Preserving Machine Learning. Utilizing Secure Multiparty Computation (SMPC) as its core, it introduces the CrypTensor, which mimics PyTorch Tensors, allowing for familiar ML operations. Designed with practicality in mind, CrypTen offers a streamlined tensor library experience akin to PyTorch. In this study, we leverage CrypTen to implement parts of the Delphi protocol for the PI evaluation of our xMLP model.

**Activation functions** provide the much-needed nonlinearity for DNNs' architecture design. While ReLU is dominant in modern DNNs, its evaluation in PI requires complex cryptographic techniques. The Gaussian Error Linear Unit (GELU), used in recent transformer models, also poses challenges in PI due to its reliance on the Gaussian kernel and $tanh$ which is not PI-friendly. Polynomial activations, compared to traditional functions like ReLU, often suffer from diminished accuracy due to issues like gradient vanishing and explosion. However, in the context of PI, they offer computational advantages, making them a preferred choice in several PI works. Furthermore, some work (Bu & Karpatne, 2021; Fan et al., 2020) explores the possibility of combining them with ReLU to enhance the expressive power of networks.

**The Vision Transformer (ViT)** applies the Transformer architecture to vision tasks by tokenizing images into fixed-size patches, embedding them as vectors, and processing them through a Transformer encoder. This allows it to capture long-range dependencies in images. As shown in Figure 2, variants like PoolFormer (Yu et al., 2022) integrate pooling with Transformers to enhance local information capture, while ResMLP (Touvron et al., 2022) offers a feed-forward approach combining residual connections and MLPs without the Transformer's attention mechanism. Meanwhile, MLP-Mixer (Tolstikhin et al., 2021) employs MLPs for both the spatial and channel dimensions of images, bypassing the need for convolutions or attention altogether. Despite their success in computer vision, these architectures have not yet been explored in PI.

## 3 THE PROPOSED XMLP FOR PI

### 3.1 INTUITIVE UNDERSTANDING OF RELU'S IMPACT

Before deep diving into the nuances of activation functions, it's imperative to understand why ReLU often outperforms polynomial activation functions in neural networks, especially in CNNs. The superiority of ReLU over polynomial activation functions is often attributed to vanishing or exploding gradient problems. However, from our in-depth studies and evaluations, we argue that ReLU's advantage indeed stems from its sparsity-inducing property, rather than polynomial activations inherently suffering from gradient problems. Recognizing this distinction is pivotal, as it offers a fresh perspective on the underlying mechanisms that underpin the success of activation functions in deep neural architectures.

ReLU zeroes out negative neuron outputs, promoting sparse features beneficial for learning (Glorot et al., 2011), as shown in Figure 3a. In contrast, the square activation keeps all outputs, lacking this sparsity feature. During the training, ReLU can selectively zero out outputs from neurons deemed less beneficial for learning. This property is vital for CNNs. Intuitively, convolution capitalizes on the spatial relationships in input data, such as pixels in images. It links each neuron only to its close neighbors

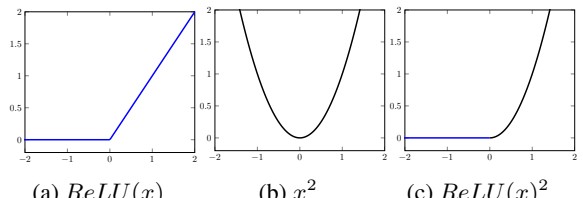

(a) $ReLU(x)$      (b) $x^2$      (c) $ReLU(x)^2$

Figure 3: $ReLU(x)^2$ has the sparsity-inducing ability but also suffers from gradient vanishing and exploding problems.

within a "receptive field". However, as the network deepens, these receptive fields expand, making neurons more globally oriented. This leads to an "information compounding" effect, wherein deep-layer neurons accumulate information from the whole input, resulting in feature maps that aren't optimal for convolution operations. As a consequence, global features can sometimes introduce redundant information, potentially degrading the overall learning outcome.

In short, *the convolutional process thrives on data with sparse connections, a condition that ReLU adeptly facilitates. On the other hand, the square function doesn't share this capacity for inducing sparsity.* To further validate our perspective, we compared the performance of $ReLU(x)$ and $ReLU(x)^2$ (as shown in Figure 3) in Section 4.2, investigating if $ReLU(x)^2$ can match ReLU's performance despite potential gradient issues. Results from Section 4.2 affirm that ReLU(x) performs on par with ReLU, underscoring our viewpoint.

### 3.2 XMLP ARCHITECTURE DESIGN

To design a network architecture for efficient PI, instead of retaining some ReLUs as in previous works, we aim to completely get rid of the ReLU function. To sidestep the aforementioned issue of "information compounding", we lean towards adopting a denser network structure. This structure shouldn't rely on local connectivity benefits like CNNs do, thereby minimizing performance loss when switching from ReLU to polynomial activations. Hence, we opt for a ViT-like architecture, where most of the linear operations are matrix multiplications rather than convolutions. Furthermore, we chose to omit the self-attention module, which isn't particularly conducive for PI.

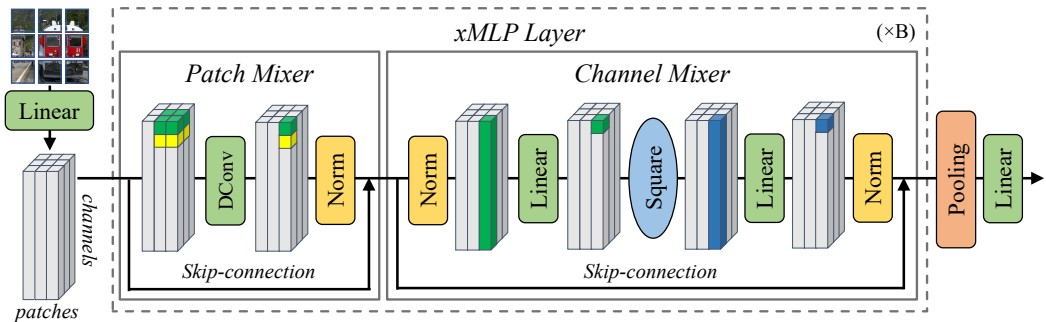

Figure 4: The xMLP architecture. xMLP consists of a patch-embedding layer, xMLP layers and a classifier head. Each xMLP layer consists of (1) a patch mixer with a residual depth-wise convolution for exchanging cross-patch information, and (2) a channel mixer layer (xMLP block) with a residual MLP with quadratic activation for exchanging cross-channel information.

xMLP takes $n \times n$ non-overlapping image patches as input and linearly projects them to $d$-dimensional embeddings while keeping the spatial arrangement of patches. This forms a 3-dimensional embedding tensor $X \in \mathbb{R}^{d \times n \times n}$. Similar to MLP-Mixer and ResMLP, the embeddings of $X$ are then fed to several repeated xMLP layers, each of which contains a residual linear patch mixer and a residual non-linear channel mixer.

The patch mixer layer in xMLP involves only dense linear operations as described by:

$$U = \text{Norm}(\text{Conv}(X)) + X, \tag{1}$$

where $X \in \mathbb{R}^{d \times n \times n}$ is the input embedding tensor, $\text{Conv}(\cdot)$ denotes the depth-wise convolution, and $\text{Norm}(\cdot)$ denotes the normalization operation as used in xMLP to be discussed below as shown in Equation (3).

The channel mixer layer in xMLP is built entirely upon two multi-layer perceptrons (MLPs) with square activation functions. The channel mixer layer can be formally described as:

$$
\begin{aligned}
V &= \text{Reshape}(\text{Norm}(U)), \\
Y &= \text{Norm}(\text{Reshape}^{-1}(W_{out}((W_{in}V) \circ (W_{in}V)))) + U,
\end{aligned}
\tag{2}
$$

where $U \in \mathbb{R}^{d \times n \times n}$ is the previous patch mixer layer's output, $W_{in}, W_{out} \in \mathbb{R}^{d \times d}$ are learnable parameters of two dense linear layers, respectively, and $\circ$ denotes the element-wise matrix multiplication. $\text{Reshape}(\cdot)$ returns the transposed of flattened patch embeddings $\mathbb{R}^{n^2 \times d}$. $\text{Reshape}^{-1}(\cdot)$ is the inverse operation of $\text{Reshape}(\cdot)$.

We apply the pre-normalization at the beginning of the patch mixer layer, and the post-normalization at the ends of both the patch mixer layer and the channel mixer layer. A pre-normalization is a patch-wise batch normalization (BN) layer without affine parameters. A post-normalization is the combination of a patch-wise batch normalization (Ioffe & Szegedy, 2015) and channel-wise scale factors. For the patch embedding $X_{i,j}$, we normalize its $k^{th}$ element by

$$\hat{X}_{i,j}^{(k)} = \frac{X_{i,j}^{(k)} - \mu_{i,j}}{\sqrt{\sigma_{i,j} + \epsilon}} * \gamma^{(k)}, \tag{3}$$

where $\mu_{i,j}, \sigma_{i,j} \in \mathbb{R}$ are the BN parameters estimated from the mean and variance of the embedding $X_{i,j}$ within a batch, respectively, both of which are computed during the training based on batch statistics, and are kept as constants during inference. $\gamma^{(k)} \in \mathbb{R}$ are the learnable affine parameters that are applied to the $k^{th}$ element of patch embeddings (shared among all embeddings).

The patch mixer layer and the channel mixer layer together form one xMLP layer, which then repeats multiple times (denoted as $\times B$) as shown in Figure 4. Finally, we use an average pooling layer to aggregate all the processed patch embeddings to a $d$-dimension vector by averaging, and then feed it into a final dense linear classifier head for the final prediction task, such as predicting the image class labels.

## 4 EXPERIMENTS

In this section, we conduct detailed experiments to evaluate the performance of xMLP on image classification tasks, CIFAR-100, Tiny-ImageNet, and ImageNet. We then perform various ablation studies to validate our hypothesis on the impacts of ReLU and square activation on accuracy. Through our experiments, we show that (1) our model, xMLP, achieves better accuracy than ResNet with fewer parameters and fewer activation layers with only square activations; (2) different from ResNet, the form of activation functions has much less impact on the accuracy of xMLP; (3) the high accuracy of xMLP models can be generalized to harder tasks such as ImageNet. In the end, we evaluate the PI latency of xMLP and show that it indeed outperforms existing SOTA PI models.

### 4.1 IMAGE CLASSIFICATION

We evaluate xMLP on image classification datasets under various configurations as shown in Table 1.

**CIFAR-100.** The dataset includes 60K 32 x 32 images in 100 classes. The dataset is split into a training set of 50K images and a test set of 10K images. We applied additional data augmentation techniques such as AutoAugment (Cubuk et al., 2018), CutMix (Yun et al., 2019) with a cut probability of 0.5, and RandomCrop. We train all models on CIFAR-100 for 200 epochs and use the AdamW optimizer (Loshchilov & Hutter, 2017) with $\beta_1$ = 0.9, $\beta_2$ = 0.99, weight decay as 0.05, a learning rate of 0.001, and a linear learning rate with WarmUp and cosine annealing scheduler. The batch size is set to 64. The training of a single xMLP-M16 model takes roughly 2 hours on two A100 GPUs.

As shown in Table 2, we first show that xMLP models are more data-efficient compared with ResMLP. We also that xMLP-T36 achieves better accuracy than ResNet-50 (78.71% versus 77.44%) with fewer parameters (10.8M versus 23.7M). The same also holds for xMLP-M24 versus ResNet-34 and xMLP-M16 versus ResNet-18. This validates our hypothesis that square activation can perform equally well (if not better) than modern CNN models so long as we design a suitable DNN architecture like xMLP.

**Tiny ImageNet.** Tiny ImageNet is the subset of ImageNet, it contains 100,000 images of 200 classes (500 for each class) downsized to 64×64 colored images. We use the same experiment

| Configurations | M | T | S |
|---|---|---|---|
| Image size | 32 | 32 | 32 |
| Patch size | 4 | 4 | 4 |
| Embedding dimension | 256 | 384 | 512 |
| Channel-mixing dimension | 256 | 384 | 2048 |
| Patch-mixing dimension | 64 | 64 | 64 |

Table 1: Model configurations.

| Models | Activation | FLOPS (G) | Top-1 (%) |
|---|---|---|---|
| ResNet-18 | ReLU | 1.1 | 75.43 |
| ResNet-34 | ReLU | 2.3 | 76.73 |
| ResNet-50 | ReLU | 2.6 | 77.44 |
| ResNet-164 | ReLU | 0.52 | 75.67 |
| ResMLP-S12 | GELU | 3.3 | 74.02 |
| ResMLP-S24 | GELU | 6.6 | 75.56 |
| ResMLP-M16 | GELU | 0.3 | 68.19 |
| ResMLP-M24 | GELU | 0.5 | 67.67 |
| xMLP-M16 (Ours) | Square | 0.4 | 75.52 |
| xMLP-M24 (Ours) | Square | 3.2 | 76.93 |
| xMLP-T36 (Ours) | Square | 3.2 | 78.71 |

Table 2: Results on CIFAR-100. We report Top-1 test accuracy on CIFAR-100. ResMLP models follow the architecture from Touvron et al. (2022) and configurations in Table 1. ResNet-18/34/50 follow the architectures from He et al. (2016a) while the first convolution layer uses the 3x3 convolution kernel, ResNet-164 is a lightweight model from He et al. (2016b).

| Dataset | Models | Activation | Top-1 (%) |
|---|---|---|---|
| Tiny-ImageNet | xMLP-M16 | Square | 63.84% |
| | xMLP-M16 | ReLU | 64.59% |
| | ResNet-18 | Square | 63.68% |
| ImageNet | xMLP(20M) | Square | 72.83% |
| | xMLP(20M) | ReLU | 70.20% |

Table 3: Results on Tiny ImageNet and ImageNet.

setting as CIFAR-100 experiments. We provide the results in Table 3. Similar to CIFAR-100 experiments, ReLU provides slightly higher accuracy to xMLP-M16 model, while xMLP-M16 with square activation has higher accuracy than ResNet-18.

**ImageNet.** On ImageNet, We trained 16-layer xMLP models (384 embedding dimension, 1536 channel-mixing dimension, and 20M parameters) on the open-source baseline Big Vision (Beyer et al., 2022) for 90 epochs. xMLP models achieve 72.83% top-1 accuracy with square activations and 70.20% with ReLU activations on the validation set. This shows that square activations consistently

provide comparable accuracy as ReLU in xMLP models even in harder tasks. Since recent all-MLP architectures such as ResMLP (Touvron et al., 2022) and MLP-Mixer (Tolstikhin et al., 2021) have already shown competence on vision tasks, we can thus confirm the generalizability of xMLP.

## 4.2 ABLATION STUDIES

**Impact of activation functions.** We perform various ablation studies to show the impact of various components as used in xMLP. Table 4 shows the ablation study on the impact of square activation. We first replace ReLU activation with square activation in ResNet. The results confirm that square activation indeed performs poorly for CNN-like networks. Interestingly, we observe that the shallower network (ResNet-10) is more tolerable with square activation than deeper networks (ResNet-18 and ResNet-34), in fact, ResNet-34 didn't even converge. This in part explains why square activation is not used in conventional DNN models. Different from ResNet, xMLP-M16 can take advantage of square activation, albeit ReLU still performs slightly better.

This is expected as ReLU still has a better opportunity to filter out the information compounding effects than square activation. To further show that our proposed xMLP architecture is indeed more resilient to the effects of information compounding, we convert the xMLP to a conventional CNN network by removing the patch mixer layer and replacing the MLP layer in the channel mixer with a 3×3 convolution layer. We denote it as *xCNN*, and we notice that, although the square activation produces lower accuracy than ReLU, the ReLU-Square activation (Figure 3c) can produce a similar accuracy as ReLU, which supports our claim about ReLU's impact on CNNs. In summary, these results show that xMLP is an efficient DNN architecture for square activation.

| Models | Top-1 (%) | | |
|---|---|---|---|
| | ReLU | Square | ReLU-Square |
| xCNN-M16* | 71.35 | 69.96 | 71.21 |
| xMLP-M16 | 76.48 | 75.52 | / |
| ResNet-10 | 70.21 | 55.86 | / |
| ResNet-18 | 75.43 | 19.51 | / |
| ResNet-34 | 76.73 | 10.00 | / |

Table 4: Comparison on the impact of activation functions on CIFAR-100. xCNN-M16* is derived from xMLP with all xMLP layers replaced by residual convolution layers. ReLU-square is the combination of ReLU and square activation function as shown in Figure 3c.

**Ablation study on the architecture rationality.** We further conduct more ablation studies as shown in Table 5 to examine the impacts of different choices of network depth, normalization functions, activation types, different operations for the patch mixer and channel mixer, and some of their combinations on xMLP's accuracy. The combination of patch-wise batch normalization and channel-wise affine produces the best accuracy over other options. Moreover, xMLP models do not need the layer normalization for convergence which is costly in PI. The activation section of Table 5 confirms the same observation as we discussed above. The patch mixer section shows that getting the global information at the patch level through either depth-wise convolution or convolution is important for xMLP's accuracy. When a dense linear operation is used for the patch mixer layer, there is a slight accuracy drop. The same observation holds for the channel mixer layer too. The combination of patch mixer layers and activation tells a similar story. Overall, these ablation studies support our hypothesis and our design of the xMLP architecture with square activation.

## 4.3 PRIVATE INFERENCE OF XMLP

We evaluate the performance of xMLP's PI under a two-party semi-honest threat model, where only one party may be compromised but still follows the PI protocol. Through our experiment results, we want to show that (1) private square activation is magnitude faster than private ReLU in existing PI systems; (2) xMLP models constantly achieve lower latency with higher accuracy compared with previous SOTA works. To the end, we showcase the performance analysis results of xMLP models and demonstrate the acceleration achieved by leveraging GPUs for computation offloading.

### 4.3.1 PI EXPERIMENT SETTINGS

Following DELPHI's protocol (Mishra et al., 2020), during the online phase of PI, as the input tensors are secret shared among the client and the server, we can process the linear layers such as

| Ablation | Model | Params | Variant | Top-1 (%) |
|---|---|---|---|---|
| Baseline models | xMLP-M16 | 2.2 | 16 layers, embedding dimension 256 | 75.52 |
| | xMLP-M24 | 3.3 | 24 layers, embedding dimension 256 | 76.93 |
| | xMLP-T36 | 10.8 | 36 layers, embedding dimension 384 | 78.71 |
| Normalization | xMLP-M16 | 2.2 | Patch-wise Affine, Patch-wise BatchNorm | 75.00 |
| | xMLP-M16 | 2.2 | Patch-wise Affine, Channel-wise BatchNorm | 74.20 |
| | xMLP-M16 | 2.2 | Patch-wise Affine, Channel-wise BatchNorm | 74.80 |
| | xMLP-M16 | 2.2 | Channel-wise Affine, Channel-wise LayerNorm | 75.17 |
| Activation | xMLP-M16 | 2.2 | square $\rightarrow$ ReLU | 76.48 |
| | xMLP-M16 | 2.2 | square $\rightarrow$ GELU | 76.43 |
| Patch mixer | xMLP-M16 | 2.2 | 3×3 dconv $\rightarrow$ none | 58.82 |
| | xMLP-M16 | 2.2 | 3×3 dconv $\rightarrow$ linear | 73.13 |
| | xMLP-M16 | 11.6 | 3×3 dconv $\rightarrow$ 3×3 conv | 77.28 |
| Channel Mixer | xMLP-M16 | 19.0 | linear $\rightarrow$ 3×3 conv | 73.63 |
| Patch Mixer & Activation | xMLP-M16 | 2.2 | 3×3 dconv $\rightarrow$ linear, square $\rightarrow$ ReLU | 73.67 |
| | xMLP-M16 | 2.2 | 3×3 dconv $\rightarrow$ linear, square $\rightarrow$ GELU | 74.49 |

Table 5: Ablation on CIFAR-100. We report top-1 accuracy on the test set of CIFAR-100. "dconv" stands for the depth-wise convolution, while "conv" stands for the standard convolution.

fully-connected layers and BatchNorm layers directly on the secret shared tensor which is as fast as plaintext linear operations. For non-linear layers, we use GC to process ReLU layers and BT to process square layers. xMLP models consist of only arithmetic operations, so its private inference can be processed efficiently through SS and BT, we report its PI latency using our PI implementation based on the open-source framework CrypTen with all the latency data collected on a system with an Intel Xeon Gold 6330 2.00GHz CPU in the LAN setting. The experiments involving GPUs are performed on A100 GPUs.

### 4.3.2 MICROBENCHMARKING

We present microbenchmarking results for single non-linear operations in Table 6, which were derived by measuring the private inference latency of processing a neural network layer and dividing the latency by the number of individual operations. We employed the open-source implementation of DELPHI (mc2 project, 2022) to measure the private ReLU latency and our implementation that based on CrypTen to measure the private square operation, and estimate the individual ReLU latency

| Private Operation | Protocol | Lat. (µs) |
|---|---|---|
| ReLU | Garbled Circuit (DELPHI) | 21 |
| | CryptFlow2* | 2 |
| Square | Beaver's Triple (CPU) | 0.05 |
| | Beaver's Triple (GPU) | 0.02 |

* Latency are measured from the reported data from Rathee et al. (2020) that the private inference of ResNet32 (with 300K ReLU) in CryptFlow2 takes 0.63s for non-linear operations, which is roughly 2µs per ReLU.

Table 6: Online latency of individual ReLU and square.

based on the data reported in CryptFlow2's paper (Rathee et al., 2020). Our results demonstrate that, despite the significant improvement (10× faster than DELPHI) in private ReLU latency achieved by the state-of-the-art private inference system CryptFlow2, private square remains orders of magnitude faster (40× on CPU, 100× on GPU). Moreover, when the computation is offloaded to GPUs, private square gains a 3.5× speedup, which is an advantage of square over ReLU since private ReLU cannot be offloaded to GPU in the existing PI systems.

### 4.3.3 RESULTS

**xMLP outperforms existing SOTA.** We provide Accuracy vs. PI latency comparisons of xMLP with previous SOTA PI works (Cho et al., 2022b; Jha et al., 2021; Cho et al., 2022a; Ghodsi et al., 2020) (latency measured in DELPHI ) in Figure 1, and the numerical results in Table 7. It is clear that xMLP's PI latency is significantly lower than all other PI models with the same level of accuracy. Figure 1 shows us that xMLP achieves the new Pareto frontier on Latency vs. Accuracy space. We compare our model with current SOTA PI model SNL (Cho et al., 2022b). For the smallest SNL model, it takes 1.07s to achieve 73.75% accuracy; in contrast, with a similar latency (1.05s), xMLP-T36 can achieve 78.71% accuracy, which is 4.96% higher. To achieve the same level of accuracy as

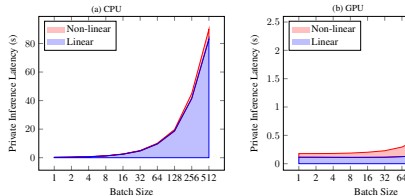

Figure 5: Online inference latency breakdown.

| Batch Size | 1 | | 32 | | 512 | |
|---|---|---|---|---|---|---|
| Operation | Linear | Non-linear | Linear | Non-linear | Linear | Non-linear |
| CPU | 0.22 | 0.04 | 4.74 | 0.29 | 83.00 | 6.99 |
| GPU | 0.11 | 0.06 | 0.12 | 0.12 | 0.25 | 2.12 |

Table 8: Online inference latency breakdown of xMLP-M16 (in seconds). PI latencies of different inference batch sizes are split into portions attributable to linear computations and non-linear computations.

the 120K ReLU SNL model (76.93% vs 76.35%), xMLP-M24 is $7\times$ faster in terms of PI latency (0.40s vs 2.80s).

**Latency source breakdown.** We break down the online inference latency of xMLP into linear latency and non-linear latency, and present the breakdown results of xMLP-M16 model on CIFAR-100 running on both GPU and CPU with different batch sizes in Figure 5. For a batch size of 1, non-linear operations account for 0.04s of xMLP-M16's PI latency, while linear operations account for 0.26s. In contrast to prior research, such as DELPHI, where non-linear operations (ReLUs) accounted for 90% of the latency, we find that non-linear operations in xMLP-M16 contribute only 16% of the total latency. This is primarily due to the faster performance of the private square compared to private ReLU. Note that, our implementation performs linear operations directly on 64-bit integer tensors, which is different from the DELPHI PI system that values are cast to float numbers, and thus produces higher overhead for the linear operation, and we believe that the latency of linear operations in our PI implementation can still be optimized.

| Method | #ReLUs (K) | Top-1 (%) | Lat. (s) |
|---|---|---|---|
| CryptoNAS (Ghodsi et al., 2020) | 100.0 | 68.50 | 2.30 |
| CryptoNAS (Ghodsi et al., 2020) | 344.0 | 76.00 | 7.50 |
| Sphynx (Cho et al., 2022a) | 102.4 | 72.90 | 2.39 |
| Sphynx (Cho et al., 2022a) | 230.0 | 74.93 | 5.12 |
| DeepReDuce (Jha et al., 2021) | 28.7 | 68.68 | 0.56 |
| DeepReDuce (Jha et al., 2021) | 49.2 | 69.50 | 1.19 |
| DeepReDuce (Jha et al., 2021) | 197.0 | 75.51 | 3.94 |
| DeepReDuce (Jha et al., 2021) | 229.4 | 76.22 | 4.61 |
| SNL[*] (Cho et al., 2022b) | 49.9 | 73.75 | 1.07 |
| SNL[*] (Cho et al., 2022b) | 120.0 | 76.35 | 2.80 |
| SNL[*] (Cho et al., 2022b) | 150.0 | 77.35 | 3.40 |
| SNL[*] (Cho et al., 2022b) | 180.0 | 77.65 | 4.05 |
| SENet[*] (Kundu et al., 2023) | 24.6 | 70.59 | 0.53 |
| SENet[*] (Kundu et al., 2023) | 49.6 | 75.28 | 1.06 |
| SENet[*] (Kundu et al., 2023) | 100 | 77.92 | 2.33 |
| SENet[*] (Kundu et al., 2023) | 150 | 78.32 | 3.50 |
| xMLP-M16 (Ours) | 0 | 75.52 | 0.27 |
| xMLP-M24 (Ours) | 0 | 76.93 | 0.40 |
| xMLP-T36 (Ours) | 0 | 78.71 | 1.05 |

[*] Latency are empirically estimated as reported in SNL paper (Cho et al., 2022b). More specifically, the latency for 1000 ReLU operations of DELPHI are measured first, which is 0.021 seconds per 1000 ReLUs, it's then used to estimate SNL networks' PI latency based on the ReLU count.

Table 7: **Comparisons on the online inference latency.** We report the top-1 accuracy on CIFAR-100 and online inference latency measure in DELPHI. "#ReLU" stands for the number of individual ReLUs in the network.

**Offloading computation to GPU.** The Beaver's Triple protocol is based solely on arithmetic matrix operations, which enables easy computation offloading to GPUs. Results in Table 8 and Figure 5 demonstrate that GPU offloading significantly reduces linear latency. It also provides a $3.3\times$ speedup for non-linear (square) operations with a batch size of 512, completing the private inference in just 2.12s, or 0.004s per image. In comparison, Table 7 shows that for PI networks requiring private ReLU operations, such as DeepReDuce's (Jha et al., 2021) ResNet-18 (with 197K ReLU), with similar accuracy (75.51% vs. 75.52%), xMLP-M16 is almost $1000\times$ faster, because private ReLU operations are limited by its bit-wise operations, and cannot be easily offloaded to GPUs (also not available in the existing PI systems) to leverages the underlying computation power like private square operations.

## 5 CONCLUSION

In this paper, we propose a novel PI-friendly DNN architecture, xMLP, which uses only square activation in the network and shows comparable efficiency and accuracy to conventional DNNs such as ResNet. xMLP demonstrates that square activation functions can be effectively used in DNN in place of ReLU activation, and xMLP's architecture allows us to replace the ReLU activation entirely with square activation without accuracy loss. Our results demonstrate xMLP achieves up to $7\times$ speedup or 4.96% higher accuracy over the state-of-the-art PI models (SNL). By demonstrating that networks constructed solely with square activations can match the performance of ReLU networks, xMLP offers a fresh avenue for addressing PI problems. Rather than tinkering with current CNN models, forthcoming research can concentrate on developing novel network structures tailored to PI challenges.

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
