# OpenReview forum: "xMLP: Revolutionizing Private Inference with Exclusive Square Activation"
_ICLR.cc/2024/Conference — Submitted to ICLR 2024_

### Official Review · Reviewer_gNjC · 2023-10-29

**Soundness:** 2 fair
**Presentation:** 3 good
**Contribution:** 2 fair
**Rating:** 5
**Confidence:** 3

**Summary:**

1. The paper proposes a novel deep neural network architecture called xMLP for Private Inference (PI) tasks. The authors identify that square activations can be processed much faster than ReLU, but using them leads to a drop in model accuracy. They attribute this accuracy drop to an "information compounding" effect.

2. PI enables deep learning models to work on private data without compromising privacy. However, existing PI systems suffer from high latency when using non-linear activations like ReLU.

3. Leveraging this insight, the authors design xMLP, which exclusively uses square activations while maintaining accuracy and efficiency comparable to ReLU-based models.

**Strengths:**

1. Design a PI-friendly architecture is important for accelerating PI. The research problem of this work is meaningful.

2. Experimental results on CIFAR-100 and ImageNet datasets demonstrate that xMLP consistently outperforms ResNet models with fewer activation layers and parameters while achieving similar performance to its ReLU-based counterparts.

3.xMLP achieves superior performance compared to state-of-the-art PI models, with a 0.58% increase in accuracy and 7x faster PI speed. Additionally, when offloading PI to the GPU, xMLP is up to 700x faster than the previous state-of-the-art PI model with comparable accuracy.

**Weaknesses:**

1. The author mentioned that they use Delphi's protocol to evaluate the proposed architecture inference. However, adding too many linear operations here would also increase the cost in offline phase. As the HE-based computation on lots of linear computation is quite costly.

2. Besides, the multiple layers here would also increase the communication cost during offline phase. As you have to do ciphertexts exchange for preparing the secret sharing masks.

**Questions:**

See Weakness part.

---

> ### Author Response · Authors · 2023-11-21
>
> Dear reviewer,
>
> Thank you for your insightful comments and the opportunity to clarify aspects of our research. We appreciate the opportunity to address your concerns and elaborate on our work.
>
> 1. Adding too many linear operations
>
> Regarding the concern that "adding too many linear operations here would also increase the cost in the offline phase," it's important to note that we use the combination of  3x3 depth-wise convolution and linear layers (matrix multiplication which is equivalent to 1x convolution). The MACs of linear operations on one xMLP block in our xMLP-M16 is 4.34M (4.2M for the fully connected layer, and 147K for the depth wise convolution layer), when compares to the standard convolution layer in ResNet18, the MACs are 18.8M, 38M, 9.4M, 2.4M in the first, second, third and forth block of ResNet. Our approach actually reduces computational requirements compared to standard convolutional. Moreover, convolution operations are not as efficient as fully connected layers in Homomorphic Encryption (HE).
>
> 2. The multiple layers would increase the communication cost during the offline phase
>
> In response to the concern that "the multiple layers here would also increase the communication cost during the offline phase," we emphasize that xMLP achieves superior results with fewer layers compared to ResNet models. Additionally, the number of layers requiring the exchange of latent features is only correlated with the number of activation layers. Consecutive linear layers do not necessitate repeated HE exchanges and can be combined in one exchange.
>
> We hope this response addresses your concerns and provides further clarity on our work. We are grateful for your feedback and look forward to any additional insights you may have.

---

> > ### Comment · Reviewer_gNjC · 2023-11-23
> > **Response to authors' rebuttal**
> >
> > Thanks for the rebuttal.
> >
> > For me, there is still a concern for using the transpose as the reshape function for intermediate tensors. As using Delphi protocol, you have to prepare the the secret-sharing masks using HE to do computation same with your online-phase. Does this cost you have considered in the evaluation? As I know, this is not as cheap as in plaintext domain due to rotation operation of HE.

---

> ### Author Response · Authors · 2023-11-23
>
> Dear reviewer,
>
> Thanks for the timely response. Rotations are only required for vector-based HE schemes such as CKKS. For other HE schemes that are not vector-based, the rotation operations are not required. And for vector-based HE schemes, rotations are also required to compute convolution layers.

---

> > ### Comment · Reviewer_gNjC · 2023-11-23
> > **Further response**
> >
> > Thanks for response.
> >
> > As I know Delphi using same HE scheme (BFV) with Gazelle (USENIX security'18). BFV and CKKS are both RLWE based HE scheme, where if you want to do position shifting for data encrypted on polynomials you can not avoid rotation operation.

---

> > > ### Author Response · Authors · 2023-11-23
> > >
> > > Yes, it's correct. The rotation operations are inevitably for reshape operations when polynomials-based HE schemes (BFV, CKKS, etc.) are used. However, the rotation operation is also required by convolution layers in all CNNs which require more rotation operations then reshape.

---

### Official Review · Reviewer_Bz3S · 2023-10-30

**Soundness:** 2 fair
**Presentation:** 3 good
**Contribution:** 2 fair
**Rating:** 3
**Confidence:** 5

**Summary:**

This paper proposes xMLP, a novel neural network architecture that uses only quadratic activations to enable fast and accurate private inference. xMLP utilizes an MLP-style architecture with global connectivity rather than convolutional layers, avoiding ReLU's sparsity-inducing effects on CNNs. xMLP combines patch and channel mixing layers with residual connections and quadratic activations.

**Strengths:**

The paper targets an important problem of enabling efficient private inference.

**Weaknesses:**

1. The analysis of why ReLU outperforms x^2 is too superficial to be a contribution. This point is not novel at all. The impact of sparsity induction and ReLU's desirable attributes have been thoroughly examined in prior works (Serra, T., Tjandraatmadja, C., & Ramalingam, S. (2018). Bounding and Counting Linear Regions of Deep Neural Networks. Proceedings of the 35th International Conference on Machine Learning.). Moreover, the comparison of relu(x), x^2, relu(x)^2 in Table 4 is not complete, making it unclear whether relu(x)^2 is better or worse that relu(x), however, which is stated in Section 3.1.

2. This paper lacks comparison with the latest methods. Authors should compare with the following papers:

a. Jha, Nandan Kumar, and Brandon Reagen. "DeepReShape: Redesigning Neural Networks for Efficient Private Inference." arXiv preprint arXiv:2304.10593 (2023).

b. Souvik Kundu, et al. Learning to linearize deep neural networks for secure and efficient private inference. International Conference on Learning Representation, 2023.

c. Kundu, Souvik, et al. "Making Models Shallow Again: Jointly Learning to Reduce Non-Linearity and Depth for Latency-Efficient Private Inference." Proceedings of the IEEE/CVF Conference on Computer Vision and Pattern Recognition. 2023.

d. Zeng, Wenxuan, et al. "MPCViT: Searching for Accurate and Efficient MPC-Friendly Vision Transformer with Heterogeneous Attention." Proceedings of the IEEE/CVF International Conference on Computer Vision. 2023.

e. Zhang, Yuke, et al. "SAL-ViT: Towards Latency Efficient Private Inference on ViT using Selective Attention Search with a Learnable Softmax Approximation." Proceedings of the IEEE/CVF International Conference on Computer Vision. 2023.

f. Dhyani, Naren, et al. "PriViT: Vision Transformers for Fast Private Inference." arXiv preprint arXiv:2310.04604 (2023).

**Questions:**

Please refer to the weaknesses.

---

> ### Author Response · Authors · 2023-11-21
>
> Dear reviewer,
>
> Thank you for your insightful comments and the opportunity to clarify aspects of our research. We appreciate the opportunity to address your concerns and elaborate on our work.
>
> We compare our results with the recent published works SENet (b. Souvik Kundu, et al. Learning to linearize deep neural networks for secure and efficient private inference. International Conference on Learning Representation, 2023).
> While SENet's comparison with SNL shows only slightly better ReLU efficiency, it's important to note that SENet still requires 50K ReLUs to achieve 75.25% accuracy, and this is close to the smallest SNL model compared in our paper which has a latency of 1.07 seconds. Our xMLP-M16 only has 0.27 seconds online latency with 75.52% accuracy. This shows our xMLP remains the advantage when compared to recent works. We will include the results that compare with more recent works in our paper.
>
> Paper [c]  uses knowledge distillation while we do not, thus we cannot directly compare with.
>
> Paper [d][e] uses 3pc protocol for latency measurements, that we cannot directly compare with since our evaluations are under 2pc setting.
>
> We hope this response addresses your concerns and provides further clarity on our work. We are grateful for your feedback and look forward to any additional insights you may have.

---

> > ### Comment · Reviewer_Bz3S · 2023-11-22
> >
> > Thanks for your response.
> > paper [d] and [e] are not necessarily under 3pc setting. They can be used under 2pc setting.
> > paper [d] replaces the softmax function with hybrid 2ReLU and Scaleattn.
> > paper [e] uses a hybrid attention architecture with self-attention and external attention, and replace softmax with learnable 2Quad function.
> > They are irrelevant to the number of parties.
> >
> > When you claim your method can be applied to ViT to reduce the costs of private inference, you need to be aware of the most recent papers in this field like [d] [e] [f].

---

> > > ### Author Response · Authors · 2023-11-22
> > >
> > > Thanks for your timely response.
> > >
> > > Experiments settings of paprer[d][e] are under 3pc, that is when they evaluate the latency, it includes the latency from the trusted third party, and their protocol for performing linear layers are completely different from ours. Even though their network can be evaluate under our settings, they didn't provide any results, and there is no proof showing that they are faster under 2pc setting because of their usage of ReLU softmax attention.
> > >
> > > Considering recent works like SENet[c], they didn't compare with ViT paper under 3pc setting as well. The comparison with paper under 3pc settings [d][e] is not reasonable.

---

### Official Review · Reviewer_9z1L · 2023-10-31

**Soundness:** 3 good
**Presentation:** 3 good
**Contribution:** 2 fair
**Rating:** 3
**Confidence:** 5

**Summary:**

This paper introduces a novel approach to enhance the efficiency of private inference (PI) by replacing ReLUs with the $x^2$. The authors employ a backbone network architecture inspired by the MLP-mixer, asserting its superior compatibility with $x^2$ compared to traditional CNNs like ResNets. Furthermore, the authors also discuss the architectural choices in terms of different combinations of ReLUs and  $x^2$, and normalization layers (BN vs LN) for optimizing the overall performance.

**Strengths:**

1. The paper introduces a novel approach by leveraging the MLP-mixer as the core network, eliminating the need for LayerNormalization for $x^2$ functioning. This is advantageous, especially for private inference where LayerNorm poses challenges.

2. Atuhros has presented a sound argument as to why substituting ReLUs with $x^2$ can be conducive to MLP-mixer-like architecture (compared to CNNs).

3. The ablation study presented in Table 5 provides valuable insights into the influence of various normalizations and the interplay between ReLU and $x^2$ on private inference performance.


4. The paper includes a comprehensive comparison of ReLU and $x^2$ on the ImageNet dataset, offering a thorough evaluation of their respective capabilities and performance characteristics.

**Weaknesses:**

$\bullet$ **Comparison with SOTA in PI:** One of the core issues with this paper is the lack of comparison with existing SOTA in PI, SENets [2]. Hence, the claim of the authors for pushing the SOTA in PI is not valid.


$\bullet$ **FLOPs comparison are not included when comparing CNN-based PI:** Another major concern with this paper is excluding the FLOPs count when comparing with CNN-based PI, in Table 2. The authors have included the params counts, which are inconclusive as the parameter counts do not signify anything in the context of PI overheads.

Since this paper, which used MLP-mixer as a backbone network, compares with the prior PI methods, which used CNN as backbone architecture, comparing only the non-linearity overheads does not tell the entire story (when comparing two distinct architecture). Moreover, the FLOPs cost cannot be ignored in PI, as they carry significant latency overhead for end-to-end latency [1]. The assumption of processing all the FLOPs offline is valid only when optimization is performed for a single inference request in isolation. However, in real-world scenarios, private inference requests arrive at non-zero rates, and processing the entire FLOPs offline becomes impractical due to limited resources. Consequently, offline costs are no longer truly offline, and FLOPs start affecting real-time performance, as illustrated in Figure 7 of the paper [1]. This effect can be exacerbated by networks with higher FLOP counts, as proposed by the authors.


$\bullet$ **Ambiguity in timing analysis:** The micro-benchmarking for timing analysis performed in Section 4.3.2 is full of ambiguity.  First, the paper said that it used a 2pc protocol, and then it included CrypTen for benchmarking. Note that Delphi and CryptFlow2 have 2pc implementation; however, CryptTen has 3pc implementation as they used TTP for generating beaver triples in the offline phase. It seems that authors have mixed 2pc and 3pc implementation for their benchmarking.

$\bullet$ **Lack of any new insight:** The discussion presented in Section 3.1 for the sparsity-inducing property of ReLU, redundancy in global features,  and "information compounding" is not novel. See [9].

$\bullet$ **Authors are ill-informed about the relevant literature:**  Authors need to tone down the claim for substituting all the ReLUs with polynomials for the first time. Prior work [6, 7, 8] replaced all the ReLUs with polynomials, and some of them demonstrated their efficacy on the ImageNet dataset too. Nonetheless, the prior work required to have higher-degree polynomials [6] or needed to use LayerNorm to mitigate the accuracy drop stemming from substituting all the ReLUs with polynomials [7].

Also, on page 9, the authors claim that implementation for offloading private computation to GPUs is not available. See the relevant work in [3,4,5].


[1] Garimella et al., "Characterizing and Optimizing End-to-End Systems for Private Inference," ASPLOS 2023.

[2] Kundu et al., "Learning to Linearize Deep Neural Networks for Secure and Efficient Private Inference," ICLR 2023.

[3] Jawalkar et al., "Orca: FSS-based Secure Training and Inference with GPUs," IEEE SP 2024

[4] Watson et al., "Piranha: A GPU Platform for Secure Computation," USENIX Security 2022

[5] Tan et al., "CRYPTGPU: Fast Privacy-Preserving Machine Learning on the GPU," IEEE SP 2021.

[6] Lee et al., "Precise approximation of convolutional neural networks for homomorphically encrypted data," IEEE Access 2023

[7]  Chrysos et al., "Regularization of polynomial networks for image recognition," CVPR 2023

[8] Xu et al., "Quadralib: A performant quadratic neural network library for architecture optimization and design exploration," MLSys 2022

[9] Zhao et al., "Rethinking ReLU to train better CNNs. " ICPR 2018


**In summary, although the approach is novel and interesting, the paper is presently in an early developmental stage and requires significant revisions.**

**Questions:**

1. Did the authors employ any knowledge distillation techniques or fine-tuning for their networks (especially in Fig. 1)?

2. Why is CrypTen used for timing analysis (in conjunction with Delphi and CrypTFlow2)?


3. How do you evaluate the runtime for depth-wise convolution?

---

> ### Author Response · Authors · 2023-11-21
>
> Dear reviewer,
>
> Thank you for your insightful comments and the opportunity to clarify aspects of our research. We appreciate the opportunity to address your concerns and elaborate on our work.
>
> 1. Comparison with Current SOTA (SENet)
>
> While SENet's comparison with SNL shows only slightly better ReLU efficiency, it's important to note that SENet still requires 50K ReLUs to achieve 75.25% accuracy, and this is close to the smallest SNL model compared in our paper which has a latency of 1.07 seconds. Our xMLP-M16 only has 0.27 seconds online latency with 75.52% accuracy. This shows our xMLP remains the advantage when compares to the recent works. We will include the results that compare with more recent works in our paper.
>
> 2. FLOPs Impact on Offline Phase
>
> The ResNet18 in our experiments have 3.4 GFLOPS and xMLP-M16 has 0.4 GFLOPs for the same input size (32x32 rgb image). We will include more comparisons of FLOPS in our paper. We also want to emphasize that during the offline phase that convolution can be slower than it is on plaintext as they require more computations due to its complexity, and they are not as optimized in HE as they are on plaintext. Therefore, we focused on comparing the end-to-end online latency.
>
> 3. Novelty in Sparsity Inducement
>
> Our novel contribution is not centered on ReLU's sparsity-inducing property. Instead, we emphasize that this property leads to suboptimal results with square activations in CNNs. Using MLP-like structures, we effectively mitigate the issues associated with square activations.
>
> 4. ill-informed about the relevant literature
>
> While these works utilized polynomial activation, they also simultaneously used ReLU functions [8],(https://github.com/zarekxu/QuadraLib/blob/main/image_classification/models/resnet.py#L83C18-L83C18). We also highlight that high-degree polynomials lead to impractical latency in PI [6]. Additionally, LayerNorm, requiring normalization operations, is unsupported in existing PI frameworks[7], even with Newton-Raphson's approximation of $1/x$, it requires at least quadratic computations. Thus, our use of simple quadratic functions and the absence of BatchNorm are significant contributions that should not be ignored. The comparison with [6][7][8] are unfair in the context of PI.
>
> 5. GPU Offloading in [3][4][5]
>
> The references [3][4][5] doesn't support for Garbled Circuits (GC) computations on GPU. Instead, they ORCA[3] and CryptGPU[5] are based on secret sharing, and Piranha[4] uses edaBits. We want to emphasize, even with recent ReLU protocols, square activations (multiplications) are still significantly faster than ReLU on both CPU and GPU. For example, as reported in CryptGPU[5], the VGG-16 online inference latency for ReLU with a batch size of 32 using CryptGPU on CIFAR dataset (32x32 image size) still takes 4.74 seconds, while the online inference of square activations in xMLP-M16 with the same batch size and image size only takes 0.29 seconds on CPU and 0.12 seconds on GPU.
>
> 6. Why use CrypTen Framework
>
> We focus on online latency, CrypTen framework use Torch for computations and can provide good performance for online inference and GPU accelerations. We have modified the CrypTen framework to perform the DELPHI's online inference. The trusted third party (TTP) in CrypTen can be seen as an alternative to offline processing, they both only need to perform once for each model and have no impact on the online inference.
>
> 7. Depth-wise Convolution
>
> Depth-wise convolutions are grouped convolutions where the number of groups is the same as the number of input channels. Like other linear layers, the computation of depth-wise convolution during online phase in our experiments is based on Torch Tensors. During the online inference, only a single linear calculation is required for the secret shared data, with the only difference being that this calculation uses LongTensors.
>
> 8.  Knowledge Distillation
>
> Our model does not involve any knowledge distillation and is entirely trained from scratch, as detailed in our paper.
>
> We hope this response addresses your concerns and provides further clarity on our work. We are grateful for your feedback and look forward to any additional insights you may have.

---

> > ### Comment · Reviewer_9z1L · 2023-11-22
> > **Reply to authors' rebuttal**
> >
> > Thanks for providing a detailed rebuttal.
> >
> > $\bullet$ Comparison with SOTA
> >
> > The improvement of SENets over SNL is beyond marginal and must be included in Figure 1 of the draft. For a better comparison, the author can employ knowledge distillation techniques to highlight the proposed methods' merits further.
> >
> > $\bullet$ FLOPs Impact on Offline Phase
> >
> > I believe ResNet18 has 0.6 GFLOPS for CIFAR-100 (32x32 RGB image). Even when comparing online latency, not the end-to-end latency, comparing FLOPs with previous methods, which used ResNet18 or WRN-22x8, is required as these backbone networks belong to two distinct classes of DNN, CNN vs MLP-Mixer.
> >
> > $\bullet$ Crypten TTP:  It is not just the question of cost (linear and non-linear operations) dynamic but also the security/threat model. When a third party is involved in generating beaver triples in the offline phase, you need to make additional assumptions about their honesty. It should be explicitly mentioned in the draft.
> >
> > Other concerns are addressed in the rebuttal.  Given the current version of the draft, I will keep my score.

---

> > > ### Author Response · Authors · 2023-11-22
> > > **Thanks for your timely response**
> > >
> > > Dear reviewer,
> > >
> > > Thanks for your timely response.
> > >
> > > I have updated comparison with SENets and comparison of FLOPS in our latest draft.
> > >
> > > For CrypTen, we would like to clarify that we are still using the DELPHI protocol, and using CrypTen framework only for performing the computation, so we are still using the same threat model as DELPHI. The triples for example, can be provided either with a TTP or HE-based algorithms, and they does not effect the measuring of online inference latency.

---

### Official Review · Reviewer_TxCL · 2023-11-04

**Soundness:** 2 fair
**Presentation:** 3 good
**Contribution:** 1 poor
**Rating:** 5
**Confidence:** 4

**Summary:**

In this paper, the authors proposed an MPC-based private inference scheme by replacing ReLU with a square function.
They argued that the proposed architecture allows us to replace the ReLU activation entirely with square activation "without accuracy loss."

**Strengths:**

In order to show that performance can be sufficiently improved by replacing it with a square function, various ablation studies were conducted on several models.

**Weaknesses:**

This paper does not contain novel results. There are already numerous research results to replace ReLU in CNN with a square function. For example, AESPA, unpublished work after DELPHI, showed that high performance can be achieved in CNN only by using the square function and some other techniques. (https://arxiv.org/pdf/2201.06699.pdf)

However, in this paper, Table 4 reports that the performance of the square function is very poor in CNNs, e.g., ResNets. This is due to the lack of sufficient surveys.

The results of this paper have very marginal novelty, such that they show that good results can be obtained with a square function in MLPmixer-type models. Also, they do not provide any information on the coefficient of used square functions.

**Questions:**

How much computation is used in the offline phase for BT to be applied? In what simulation environments and how much time does it take?
The auxiliary random triples required for computation must be distributed in advance, but how much data should be distributed in advance?

**Details Of Ethics Concerns:**

I have no concern.

---

> ### Author Response · Authors · 2023-11-21
>
> Dear reviewer,
>
> Thank you for your insightful comments and the opportunity to clarify aspects of our research. We appreciate the opportunity to address your concerns and elaborate on our work.
>
> 1. Novelty and Comparison with AESPA's Methodology
>
> Firstly, it is crucial to highlight that in the AESPA report, only ResNet18 achieved a higher accuracy using their substitution method. However, for deeper networks like ResNet32 on CIFAR100, the accuracy decreased by 7.68%. This indicates the limitations of the method mentioned in their paper for deeper networks, and the results are also contradictory to its performance on CIFAR10, we doubt these results are reasonable. Despite this, our proposed xMLP model achieved accuracy comparable to ReLU across 16/24/36 layers, demonstrating its effectiveness even in deeper network architectures. We also emphasize that our contribution lies in the fact that instead of simply assuming quadratic activation functions are insufficient, we believe that the primary reason quadratic functions do not perform well in ResNets is due to limitations in the network architecture.
>
> 2. Effectiveness of the Square Function in xMLP
>
> Our research uniquely demonstrates that a simple square function, $y=x^2$ (the coefficient for the quadratic term is 1, for the linear term is 0, and the constant term also has a coefficient of 0.), and it achieves results close to those with ReLU.
>
> 3. Computation Cost for BT in the offline phase
>
> While this is not the main focus of our comparison, it's noteworthy that in systems like DELPHI using HE, the offline overhead for BT is significantly lower than the communication delay and data transfer required for ReLU (Garbled Circuits) – approximately 5% and 1%, respectively. Comparing the similar accuracy of xMLP-16 and ResNet18, the operations for square and ReLU are 262K and 311K, respectively. xMLP has less total amount BT operations and the cost for a single BT operation is significantly lower than ReLU, and for the total operations (linear and nonlinear), the ResNet18 in our experiments have 3.4 GFLOPS and xMLP-M16 has 0.4 GFLOPS, resulting in a considerably faster overall performance for the offline phase.
>
> We hope this response addresses your concerns and provides further clarity on our work. We are grateful for your feedback and look forward to any additional insights you may have.

---

### Meta-Review · Area_Chair_5iC3 · 2023-12-06

**Metareview:**

The main contribution of this paper is a neural network architecture called xMLP that only comprises square activations; such an architecture is particularly well suited for privacy-preserving inference using MPC and homomorphic encryption.

While the problem formulation is important and the approach seems reasonable, several issues were raised during the review process. The primary concern was a relative lack of novelty (there is a large body literature on replacing ReLUs with polynomial activations, which the authors don't sufficiently engage with). Other concerns included issues with evaluation (the use of CrypTen in the benchmarking experiments was questionable) and questions surrounding offline costs.

Final recommendation: reject.

**Justification For Why Not Higher Score:**

Scores from the reviewers were all mixed-to-negative (and I agree with their assessment.)

**Justification For Why Not Lower Score:**

N/A

---

### Decision · Program_Chairs · 2024-01-16

Reject